# DiffuSETS: 12-lead ECG Generation Conditioned on Clinical Text Reports and Patient-Specific Information

Jiabo Chen[*]
jiabo.david.chen@gmail.com
VCIP, CS, Nankai University
Tianjin, China

Yongfan Lai[*]
laiyf@stu.pku.edu.cn
School of Intelligence Science and
Technology, Peking University
Beijing, China

Deyun Zhang
zhangdeyun@heartvoice.com.cn
HeartVoice Medical Technology
Hefei, Anhui, China

Yue Wang
wangyue@heartvoice.com.cn
HeartVoice Medical Technology
Hefei, Anhui, China

Shijia Geng
gengshijia@heartvoice.com.cn
HeartVoice Medical Technology
Hefei, Anhui, China

Hongyan Li
leehy@pku.edu.cn
School of Intelligence Science and
Technology, Peking University
Beijing, China

Shenda Hong[†]
hongshenda@pku.edu.cn
National Institute of Health Data
Science, Peking University
Beijing, China

## ABSTRACT

Heart disease poses a serious threat to human health. As a non-invasive diagnostic tool, the electrocardiogram (ECG) is one of the most commonly used methods for cardiac screening. Obtaining a large number of real ECG samples often entails high costs, and releasing hospital data also necessitates consideration of patient privacy. Due to the shortage of medical resources, precisely annotated ECG data are scarce. In the critical task of generating ECGs, work on generating ECGs from text is extremely rare. Given the differing data modalities, incorporating patient-specific information into the generation process is also challenging. To address these challenges, we propose DiffuSETS, the first method to use a diffusion model architecture for text-to-ECG. Our method can accept various modalities of clinical text reports and patient-specific information as inputs and generates ECGs with high semantic alignment and fidelity. In response to the lack of benchmarking methods in the ECG generation field, we also propose a comprehensive evaluation method to test the effectiveness of ECG generation. Our model achieve excellent results in tests, further proving its superiority in the task of text-to-ECG. Our code and trained models will be released after the acceptance of our paper.

## KEYWORDS

Cardiology, Electrocardiogram, Signal processing, ECG generation, Diffusion models

---

[*]Both authors contributed equally to this research.
[†]Corresponding Author.

---

*KDD-AIDSH '24, August 25–29, 2024, Barcelona, Spain*
2024. ACM ISBN 978-x-xxxx-xxxx-x/YY/MM
https://doi.org/10.1145/nnnnnnn.nnnnnnn

**ACM Reference Format:**
Jiabo Chen, Yongfan Lai, Deyun Zhang, Yue Wang, Shijia Geng, Hongyan Li, and Shenda Hong. 2024. DiffuSETS: 12-lead ECG Generation Conditioned on Clinical Text Reports and Patient-Specific Information. In *Proceedings of (KDD-AIDSH '24)*. ACM, New York, NY, USA, 15 pages. https://doi.org/10.1145/nnnnnnn.nnnnnnn

## 1 INTRODUCTION

The electrocardiogram (ECG) is a non-invasive diagnostic tool for heart disease and is widely used in clinical practice [Holst et al. 1999]. Many studies have focused on developing ECG classifiers [Bian et al. 2022; Golany et al. 2021; Kiranyaz et al. 2015] and using them for automated ECG analysis. However, due to patient privacy concerns [Hazra and Byun 2020; Hossain et al. 2021], acquiring and sharing real ECG signals is a huge challenge. Accurately labeled ECG signals are also rare [Golany et al. 2020b], and obtaining them is costly [Chen et al. 2022]. Considering these issues, a key upstream task is to generate ECG signals [Golany et al. 2020b; Zhang and Babaeizadeh 2021].

In the field of ECG signal generation, the main research goal is to generate ECG signal samples with high fidelity and rich diversity. Many studies have adopted the Generative Adversarial Network (GAN) architecture to generate ECG signals [Adib et al. 2021], and others have introduced Ordinary Differential Equation (ODE) systems representing cardiac dynamics to create ECG samples [Golany et al. 2020b]. Recently, some new studies have incorporated patient-specific cardiac disease information into the ECG generation process to improve the generated outcomes [Alcaraz and Strodthoff 2023; Chen et al. 2022], and some have included the content of patients' clinical text reports during generation [Chung et al. 2023]. However, current research in this field still has certain deficiencies: **(1) Dataset limitations.** Many datasets contain a limited variety of features and lack comprehensive patient-specific

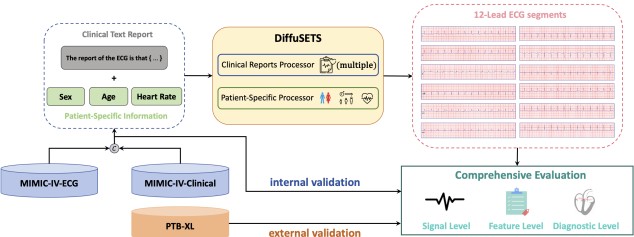

**Figure 1: High level overview of our work.**

information. Several datasets (such as PTB-XL) have low textual richness in clinical text reports. Using MIMIC-IV-ECG [Gow et al. 2023] and MIMIC-IV-Clinical [Johnson et al. 2023] requires clinical knowledge, which poses a challenge for algorithmic researchers. **(2) Difficulty in unifying features across different modalities.** Clinical text reports and patient-specific information contain data in various modalities, each with different data distributions. During the algorithmic processing, the corresponding feature vectors are also different in dimensionality. Integrating all this information into the model is challenging. **(3) Lack of benchmarking methods.** In the field of ECG generation, there is still a lack of comprehensive benchmarking methods, making it difficult to assess the relative merits of models.

To address the aforementioned issues, this paper introduces DiffuSETS, a **diffu**sion model to **s**ynthesize 12-lead **E**CGs conditioned on clinical **t**ext reports and patient-**s**pecific information. High level overview of our model as shown in Figure 1. Our approach is **the first work** to use diffusion models to handle ECGs generation from text. DiffuSETS utilizes the MIMIC-IV-ECG dataset as the training dataset, which features a wide variety of characteristics suitable for ECG signal generation and enhances the diversity of the generated signal samples. We also design a comprehensive evaluation, which includes quantitative and qualitative analyses at the signal level, feature level, and diagnostic level. Such testing allows for a comprehensive evaluation of the performance of generative models. We have also incorporated a clinical Turing test in this comprehensive evaluation, involving evaluations by cardiologists, to ensure the high fidelity of the generated ECG samples.

The main contributions of this paper are as follows: **(A)** We have integrated MIMIC-IV-ECG and MIMIC-IV-Clinical, focusing on clinical knowledge during the processing. We have utilized these datasets in the training and inference processes of our algorithms, enabling the model to accept various modalities of different features as inputs. This also provides more possibilities for future expansion. **(B)** We introduce DiffuSETS, an ECG signal generator that accepts clinical text reports and patient-specific information as inputs. This is the first work to use diffusion models for generating ECGs from text. ECG signal samples with high semantic alignment can be generated by just inputting simple natural language text as a description of the patient's disease information. We can also accept inputs such as heart rate, sex, and age, adding constraints to the features of the generated ECG signals, thus making the generation of ECG signals more detailed and diverse. **(C)** We have designed a set of comprehensive evaluation to evaluate the effectiveness of ECG signal generation, which can comprehensively assess the

performance of generative models. Our method was tested within this comprehensive evaluation, and the results were very significant, demonstrating the fidelity and semantic alignment of the model-generated ECG signal samples.

## 2 RELATED WORK

There are many studies currently attempting to address the generation of electrocardiogram (ECG) signals, but these methods have several limitations. Firstly, many models can only generate short-term time series [Delaney et al. 2019; Golany et al. 2020a; Li et al. 2022; Yoon et al. 2019], enabling them to produce only the content of a single heartbeat, rather than long-term ECG recordings. Secondly, they are often trained on small datasets with a limited number of patients [Zhu et al. 2019], or they use only a limited set of conditional labels [Golany and Radinsky 2019; Golany et al. 2020b; Sang et al. 2022]. In addition, many of these methods require ECG segmentation as a pre-training step, rather than directly processing continuous signals [Golany et al. 2020a,b; Li et al. 2022; Sang et al. 2022]. Moreover, many of these methods are capable of generating and classifying for specific patients only [Golany and Radinsky 2019], and lack comprehensive training data and samples aimed at the general population [Thambawita et al. 2021].

In recent studies, some researchers have attempted to apply diffusion models to the generation of ECGs [Adib et al. 2023], treating ECGs as images rather than time series, and their methods were limited to the unconditional generation of single-lead ECGs. Moreover, from a quantitative performance evaluation perspective, these methods have not surpassed those based on GANs for generating ECGs. ME-GAN [Chen et al. 2022] introduces a disease-aware generative adversarial network for multi-view ECG synthesis, focusing on how to appropriately inject cardiac disease information into the generation process and maintain the correct sequence between views. However, their approach does not consider text input, and therefore cannot incorporate information from clinical text reports. Auto-TTE [Chung et al. 2023] proposed a conditional generative model that can produce ECGs from clinical text reports, but they also segmented the ECGs as a preprocessing step. SSSD-ECG [Alcaraz and Strodthoff 2023] introduced a conditional generative model of ECGs with a structured state space, encoding labels for 71 diseases and incorporating them into the model training as conditions, but it cannot accept clinical text reports in the form of natural language text, thus lacking some of the rich semantic information inherent in disease diagnosis. At the same time, due to the lack of a unified performance evaluation setup, it is often challenging to quickly assess the relative merits of these methods.

## 3 METHOD

The architecture of DiffuSETS is illustrated in the Figure 2, involving three modalities: signal space, latent space, and conditional information space (clinical text reports and patient-specific information).

### 3.1 Model Architecture

The network architecture of DiffuSETS comprises a training phase and an inference phase, as depicted in Figure 2. **In the training phase,** we first extract 12-lead ECG signal $x$ from the ECG dataset.

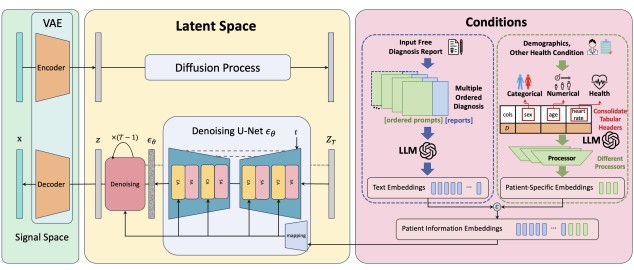

**Figure 2: Overall architecture of DiffuSETS architecture, which consists of four components: variational autoencoder, semantic embedding model, denoising diffusion process, noise prediction model.**

The signal-space representations of 12-lead ECG is then compressed by the encoder $E_\phi$ of variational autoencoder [Kingma and Welling 2013] to obtain latent-space representation of the ECGs, marked as $z_0$. Corresponding clinical text reports, after processing with prompts and utilizing an LLM, are transformed into a text embedding vector. Patient-specific information is also processed into a patient-specific embedding vector and merged with the text embedding vector to form a condition embedding vector $c$, which is then incorporated into the model's training. Subsequently, the denoising diffusion probabilistic model (DDPM, Ho et al. 2020) scheduler continuously adds Gaussian noise $\epsilon_t$ to get the latent-space representation $z_t$ at randomly sampled time step $t$ through forward process formula: $z_t = \sqrt{\bar{\alpha}_t} z_0 + \sqrt{1 - \bar{\alpha}_t} \epsilon_t$ , $\epsilon \sim \mathcal{N}(0, \mathbf{I})$ .

The noise predictor, fed with the noisy latent-space representation $z_t$, current time step $t$ and the condition embedding vector $c$, is trained to predict that noise. The loss function of the training phase is defined as:

$$\mathcal{L}_{\text{DiffuSETS}} = \|\epsilon_t - \hat{\epsilon}_\theta(z_t, t, c)\|_2^2 \tag{1}$$

where $\hat{\epsilon}_\theta(z_t, t, c)$ stands for the output of noise prediction model. By performing gradient descend on Equation 1, we can raise the Evidence Lower BOund (ELBO) so as to maximize the log likelihood of the training samples [Ho et al. 2020].

**In the inference phase,** the initial ECG signal latent $z_T$ is a noise vector sampled from the standard normal distribution. At each point during time step descends from $T$ to 1, the noise prediction model attempts to predict a noise $\hat{\epsilon}_\theta(z_t, t, c)$ with the assistance of the input clinical text reports and patient-specific information and the denoising diffusion probabilistic model scheduler denoises the latent-space representation $z_t$ to retrieve $z_{t-1}$ through a sampling process:

$$z_{t-1} \sim \mathcal{N}(\mu_q, \ \sigma_t^2 \mathbf{I}) \tag{2}$$

$$\mu_q := \left[\sqrt{\alpha_t}(1 - \bar{\alpha}_{t-1}) z_t + \sqrt{\bar{\alpha}_{t-1}}(1 - \alpha_t)\hat{z}_0\right]/(1 - \bar{\alpha}_t) \tag{3}$$

$$\hat{z}_0 := \left[z_t - \sqrt{1 - \bar{\alpha}_t}\hat{\epsilon}_\theta(z_t, t, c)\right]/\sqrt{\bar{\alpha}_t} \tag{4}$$

$$\sigma_t^2 := (1 - \alpha_t)(1 - \bar{\alpha}_{t-1})/(1 - \bar{\alpha}_t) \tag{5}$$

where $\alpha_t$ is the hyperparameter related to diffusion forward process noise. Finally, our trained decoder $D_\theta$ reconstructs the normal 12-lead ECG signal based on the denoised latent-space representation,

producing a signal-space ECG waveform series that aligns with the input descriptions.

## 3.2 Processing Clinical Text Reports

To achieve better semantic alignment with clinical text reports and patient-specific information, we designed different processing methods of conditions based on the diverse data types and distributions. The results are then merged into an embedding vector to represent the patient's features. To enhance the model's ability to accept clinical text reports in natural language format as input, we also devised prompts for these texts and utilized the semantic embedding model "text-embedding-ada-002" provided by OpenAI (referred to as ada v2).

The processing workflow for clinical text reports in this paper is shown in Figure 7. We employed a pretrained language model to process the clinical text reports. Specifically, for handling natural language text in clinical text reports, we use ada v2 to generate text embedding vectors. Before inputting the clinical text reports into ada v2, we designed prompts for processing. If only one report is inputted, the prompt is **"The report of the ECG is that {text}."** However, it is common for the dataset tables to show that one ECG corresponds to **multiple** clinical text reports, for which we have made special arrangements. In clinical datasets, the presence of multiple clinical text reports often serves to complement each other; typically, the most important report is placed first, with the remaining content supplementing the first report from various perspectives. Therefore, we designed specific ordered prompts for them. For the first clinical text report, our prompt is **"Most importantly, The 1st diagnosis is {text}."** For the subsequent reports, our prompt is **"As a supplementary condition, the 2nd/3rd/... diagnosis is {text}."** This enables the model to recognize the differences when processing multiple clinical text reports, thereby better understanding the semantic information contained within the reports.

## 3.3 Processing Patient-Specific Information

In the MIMIC-IV-ECG and MIMIC-IV-Clinical datasets, there is a wealth of tabular data recording patient-specific information. We categorize these characteristics into three types: categorical demographic condition, numerical demographic condition, and other health condition. We have designed specific processing methods for each type of data and used a large language model (LLM) to process the tabular headers in the tabular data, consolidating their information into a patient-specific embedding. Then, we concatenate the processed patient-specific embedding with the text embedding vector generated from the clinical text reports to obtain the conditions embedding vector $c$ (Using sex, age, and heart rate as examples, see Equation 6). This vector is used for both model training and inference, facilitating the model's understanding of the semantic information included in the input. The above content is illustrated in the Figure 8.

$$c = \text{Concat}\left(\text{ada\_v2}(text), \ hr, \ age, \ G(sex)\right), \ G(x) = \begin{cases} 0, & x = \text{F} \\ 1, & x = \text{M} \end{cases} \tag{6}$$

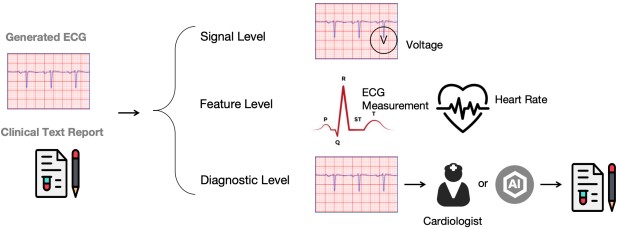

**Figure 3: The design of evaluation of ECG generation at signal level, feature level and diagnostic level.**

## 3.4 Comprehensive Evaluation of ECG Generation

Our performance metrics involves experiments and analysis at the signal level, feature level, and diagnostic level. **At the signal level,** we focus on the stability of the generated signals. We compute the error between the generated and original signals to assess the effectiveness of the ECG signal generation. **At the feature level,** we examine whether the ECG signals generated by the model align with the input descriptions of patient-specific information. Considering that sex and age are difficult to measure through quantitative analysis, we have chosen heart rate as the focal point for testing. **At the diagnostic level,** we assess whether the generated ECGs conform to the descriptions of the disease, that is, the content of the clinical text reports. We use a pretrained model to compare the generated ECG signals with the clinical text reports. This pretrained model can be viewed as a text-ECG encoder that has already achieved semantic alignment, acting like a classifier to measure whether the outcomes generated by our model match the disease conditions. Our design is shown in Figure 3.

## 4 EXPERIMENTS

### 4.1 Experimental Setup

In setting up the dataset, we use the MIMIC-IV-ECG dataset [Gow et al. 2023] to train the DiffuSETS model. MIMIC-IV project covers hospital admission records of 299, 712 patients from 2008 to 2019 at the Beth Israel Deaconess Medical Center, including patient personal information such as age and sex. The ECG dataset within it contains 800, 035 records with ECG signals, patient IDs, RR intervals, and machine-generated clinical text reports. For each signal, we search the sex and age characteristics of the ECG owners in MIMIC-IV-Clinical dataset [Johnson et al. 2023] by the patient IDs, and calculate the heart rate using the RR intervals. However, some RR intervals showed anomalies, such as 0 ms or 65, 535 ms. Therefore, for data samples where the RR intervals fall outside the range of 300 ms to 1, 500 ms, we use the XQRS detector from the wfdb toolkit [Sharma and Kohli 2023] to obtain the QRS intervals through waveform analysis to calculate the heart rate. Samples that could not calculate a heart rate from all 12 waveforms are considered to have abnormal heart rate records and are discarded along with samples missing sex or age information. After the preprocessing, we retain 794, 372 records. Each lead's original data is a 10-second ECG signal at a sampling rate of 500 Hz, resulting in 5, 000 time samples. We down-sample these to 1, 024 time samples for model training

**Table 1: Signal level results: MAE of generated and reference ECG. Random means randomly sample neglecting the clinical text reports.**

| DiffuSETS Model | MIMIC-IV-ECG | | | | | PTB-XL |
| --- | --- | --- | --- | --- | --- | --- |
| | Sinus rhythm | Sinus bradycardia | Sinus tachycardia | Abnormal ECG | Random | Random |
| DiffuSETS | 0.0864 | 0.0697 | 0.0861 | 0.0863 | 0.0908 | 0.0990 |
| w/o PS-info | 0.0898 | 0.0807 | 0.0978 | 0.0991 | 0.0910 | 0.1019 |
| w/o VAE | 0.0886 | 0.0704 | 0.1144 | 0.1002 | 0.0942 | 0.1048 |
| w/o PS-info&VAE | 0.0949 | 0.0806 | 0.1240 | 0.1196 | 0.1101 | 0.1077 |

DiffuSETS: Default DiffuSETS model
w/o PS-info: Without Patient-specific Information
w/o VAE: Without VAE Encoder-Decoder
w/o PS-info&VAE: Without VAE Encoder-Decoder and Patient-specific Information.

and internal validation. We also use the PTB-XL dataset for external validation, which contains 21, 799 clinical entries, each with a 10-second ECG signal, along with patient-specific information and doctor-recorded ECG reports. The PTB-XL labels do not include records of heart rate, so we directly use the waveforms to calculate the heart rate. Similarly, we sample the ECGs at a rate of 500 Hz and down-sample them to 1, 024 time samples, following the processing method used for MIMIC-IV-ECG dataset.

Our method is trained on a GeForce RTX 3090 using PyTorch 2.1. Batch size is set to 512, with a learning rate of $5 \times 10^{-4}$. The latent space is set to $\mathbb{R}^{4 \times 128}$. The number of time step $T$ in training phase is set to 1, 000 while noise $\beta_t$ of diffusion forward process are assigned to linear intervals of $[0.00085, 0.0120]$. Noise predictor has 5 layers and the kernel size of convolution is 7. It iterates approximate 60 time steps per second within the same environment in inference phase.

In the setup of our control experiments, due to the scarcity of open sourced pre-existing deep generative models for ECGs, we set up several variations of our DiffuSETS model for comparison and ablation study. These variations allow us to assess the performance improvements brought by each module in DiffuSETS, as well as to more comprehensively validate the rationality of the settings in each module of our method. Detailed descriptions can be found in Section 4.5.

### 4.2 Performance Metrics

*4.2.1 Signal Level.* At the signal level, we compute the stability of the generated ECG signals. We directly input the conditions extracted from the real samples of ECG dataset into the model, and then test the Mean Absolute Error (MAE) between the ECGs output by the model and the original ECGs to measure the stability of the generated signals. The results are categorized by clinical text report and are displayed in the Table 1. It can be observed that the MAE value between the 12-lead ECGs generated by our model and the original ECGs of the ground truth is very small. This indicates that when our model receives ECG signals as input, the results deviate minimally from the original ECGs, demonstrating the stability of our method.

*4.2.2 Feature Level.* At the feature level, we track the heart rate consistency between generated ECG and input feature. Our approach involves using the condition in ground truth sample as the input and obtaining heart rate value of generated ECG. The scatter

**Table 2: Diagnostic level results.**

| DiffuSETS Model* | CLIP Score | Clip Score Ground Truth | rCLIP Score | rFID Score | Precision | Recall | F1 Score |
|---|---|---|---|---|---|---|---|
| DiffuSETS | **0.38** | 0.45 | **0.84** | **0.48** | 0.97 | **0.72** | **0.83** |
| w/o PS-info | 0.37 | 0.45 | 0.81 | 0.87 | **0.99** | 0.60 | 0.75 |
| w/o VAE | 0.33 | 0.45 | 0.74 | 1.31 | 0.93 | 0.68 | 0.79 |
| w/o PS-info&VAE | 0.32 | 0.45 | 0.71 | 2.23 | 0.83 | 0.70 | 0.76 |

\* Code names of DiffuSETS models are the same with those in Table 1

plot of this experiment is depicted in Figure 4 right and the MAE result is recorded at Figure 4 left. It is suggested that patient-specific information significantly reduces the heart rate deviation, which demonstrates that our model can generate conditional ECGs finely based on the heart rate information contained in the patient-specific information. Besides, in the generation of conditional ECGs, the addition of heart rate information also makes the generated ECGs more consistent with the characteristics of the disease.

| DiffuSETS Model* | Real Condition (MAE) | |
|---|---|---|
| | MIMIC-IV-ECG | PTB-XL |
| DiffuSETS | 1.75 | 5.56 |
| w/o PS-info | 8.92 | 12.54 |
| w/o VAE | 5.06 | 11.96 |
| w/o PS-info&VAE | 15.11 | 13.51 |

\* Code names of DiffuSETS models are the same with those in Table 1

**(a) MAE**

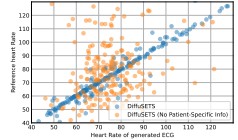

**(b) Scatters**

**Figure 4: Feature level results of generated ECG heart rate and reference heart rate.**

*4.2.3 Diagnostic Level.* At the diagnostic level, we use CLIP scores [Hessel et al. 2021] to assess the semantic alignment between the ECG signals generated by the model and the input features. Using the Contrastive Language-Image Pre-Training (CLIP), we modify its image encoder into an ECG encoder, implementing it with a method similar to that in METS [Li et al. 2024], using Resnet1d-18 [He et al. 2016; Hong et al. 2020] as the backbone network. To mitigate errors inherent in the CLIP model itself, we evaluate the CLIP score with the original ECG as input and divide the CLIP score obtained from the generated data by the ground truth CLIP score to yield a relative CLIP score (rCLIP score). Additionally, We generate feature vectors for both the ground truth and the generated ECG signals from the test set using the pre-trained model, which are then used for FID score assessment. In calculating the FID score, we also test the ground truth FID value and calculate the rFID score using the same method as for the rCLIP scores. Furthermore, we compute the precision and recall score following the steps of [Kynkäänniemi et al. 2019]. The results of this Section are all shown in Table 2.

## 4.3 Expert Evaluation

We conducted two kinds of cardiologist evaluation test. **In the fidelity evaluation**, we extracted feature information from 50 records in the MIMIC-IV-ECG dataset, used them as input to generate 50 ECGs, and randomly selected another 50 ECGs from the MIMIC-IV-ECG dataset. We provided these generated data alongside the real data to cardiologists for Turing test assessment. The cardiologists were tasked with determining whether the provided

**Table 3: Expert evaluation on fidelity and semantic alignment of generated ECG.**

| Label Judgment | Fidelity Test (Turing Test) | | | | Alignment Test |
|---|---|---|---|---|---|
| | Real ECG | | Generated ECG | | ACC |
| | Real (TP) | Generated (FN) | Real (FP) | Generated (TN) | |
| Expert 1 | 48 | 2 | 31 | 19 | 63% |
| Expert 2 | 47 | 3 | 30 | 20 | 59% |
| Expert 3 | 46 | 4 | 49 | 1 | 47% |

ECGs were generated by a machine. **In the test for semantic alignment**, we provided 100 generated ECGs (using different conditions, especially clinical text report, recorded in MIMIC-IV-ECG). Experts were asked to determine whether our generated results matched the descriptions in the clinical text reports. The results are recorded in Table 3. From the perspective of the high FP value in Turing test, the majority of generated ECGs successfully deceive the cardiology experts, thus are comparable to real ones. In alignment test, more than half of generated ECGs are considered matching to input clinical text reports, which indicates that DiffuSETS model do grasp the diagnostic information between signal and text modalities when expanding the diversity of generated ECGs.

## 4.4 Case Study

As an extension to the semantic alignment test in expert evaluation, here we post a picture of generated ECG and explain why it coordinates with the input clinical text reports. In Appendix D, we introduce more case studies and further prove that our model is compatible with generating semantic aligned ECGs from the general waveform perspective down to the single beat view.

*Atrial fibrillation accompanied by slow ventricular response and paroxysmal ectopic ventricular rhythm* is a relatively rare complex cardiac arrhythmia, and our method accurately generated this complex ECG, as shown in Figure 5. First, the patient is an atrial fibrillation sufferer, primarily characterized by irregular RR intervals and the absence of P waves, both of which are consistently represented in the generated ECG (marked as red rectangle). Secondly, the patient's ventricular rate is slow, and the generated ECG showed a ventricular rate of 56 beats per minute, matching the textual description. This is particularly challenging since most atrial fibrillation patients have a fast heart rate, and slow ventricular atrial fibrillation is rare in the training dataset. This accuracy is largely due to the method of integrating patient-specific information proposed in this paper. Finally, the generated ECG displayed broad, abnormal QRS waves (marked as blue rectangle), consistent with descriptions of paroxysmal ectopic ventricular rhythms. Overall, the consistency across multiple leads of the generated ECG is very good, both in terms of waveform alignment and the direction of the QRS main wave, aligning well with the actual conditions.

## 4.5 Ablation Study

We conducted ablation study on DiffuSETS and evaluated them using the methods described in Section 4.2, with the related performance data presented in the Tables 1 2 4. Among them, "Without VAE Encoder-Decoder" refers to the absence of the VAE encoder-decoder during training and inference, using original ECG signals

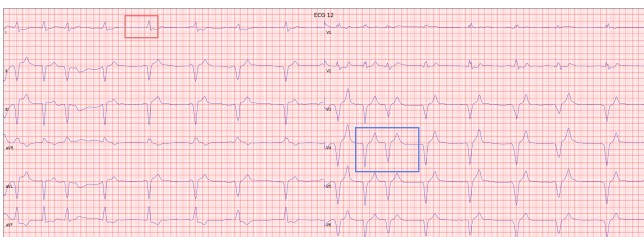

**Figure 5: The electrocardiogram generated based on the clinical text report "atrial fibrillation with slow ventricular response with paroxysmal idioventricular rhythm."**

as intermediate data in the DDPM module of DiffuSETS. "Without Patient-specific Information" indicates that the training and inference processes used only clinical text reports as input, without incorporating any additional patient-specific information for inference. The experimental results from the table above demonstrate that incorporating a VAE encoder-decoder significantly enhances the signal-level performance of the generated results. Moreover, including patient-specific information during the generation process slightly improves the outcomes at both the signal and diagnostic levels, with substantial improvements at the feature level. These results confirm that the modular design used in our method is highly effective and well-suited for the task of generating ECGs.

## 5 CONCLUSION

We propose a novel electrocardiogram (ECG) generative model, DiffuSETS, which integrates clinical text reports and patient-specific information to generate ECGs with high fidelity and semantic alignment. Additionally, we present a comprehensive evaluation for assessing generative models of ECG signals, allowing for a thorough evaluation of their performance and the fidelity and semantic alignment of generated samples.

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

# A  DETAILED STRUCTURE OF DIFFUSETS

## A.1  Variational Autoencoder

The variational autoencoder [Kingma and Welling 2013] consists of two parts: a encoder $E_\phi$ to compute the mean and variance of latent normal distribution of input ECG signal $x$ and a decoder $D_\theta$ to reconstruct the latent vector $z$ back to ECG signal. The latent-space representation is computed through reparameterization method (Equation 7) to enable the gradient pass through the discrete sampling process.

$$z \sim \mathcal{N}(\mu, \sigma^2) \iff z = \mu + \sigma \times \epsilon, \; \epsilon \sim \mathcal{N}(0, \mathbf{I}) \tag{7}$$

We train the variational autoencoder separately and whose loss function comprises two parts: reconstruction error and KL divergence. The reconstruction error uses Mean Squared Error (MSE) to measure the difference between the input ECG and the reconstructed ECG, while the KL divergence measures the difference between the encoded latent distribution and the standard normal distribution $N(0, 1)$. Combining these two parts, our loss function expression is:

$$\mathcal{L}_{\text{vae}} = \text{MSE}(x_{\text{input}}, \; x_{\text{recons}}) + \lambda \cdot D_{KL}\left(q_\phi(z|x) \; \| \; \mathcal{N}(0, \mathbf{I})\right) \tag{8}$$

$$= \frac{1}{N}\sum_{i=1}^{N}(x_i - D_\theta(z_i))^2 - \frac{\lambda}{2}\sum_{j=1}^{N}(1 + \log(\sigma_j) - \mu_j^2 - \sigma_j^2) \tag{9}$$

where $q_\phi(z|x)$ is the latent-space variable distribution and $\mu_j, \sigma_j$ are the outputs of encoder $E_\phi$. In order to alleviate the KL vanishing problem [Bowman et al. 2015], we adopt the monotonic KL-annealing where coefficient $\lambda$ starts at 0 and increases linearly with the growth of epochs.

## A.2  Noise Prediction Model

Our noise predictor follows the architecture of U-Net [Ronneberger et al. 2015], which contains a group of down-sampling layers $D_i$, a group of up-sampling layers $U_j$ and a bottleneck block concatenating two groups. The detailed architecture of noise predictor model are shown in Figure 6. Passing through a down-sampling layer, the latent vector $z \in \mathbb{R}^{C \times L}$ would be enriched in channel dimension while be shortened in length dimension and *vice versa*. Besides the directly information flow from anterior layer $U_i$ to subsequent layer $U_{i+1}$ within the up-sampling groups, there also exist skip connections linking the down-sampling layer at the same level. Therefore the input expression of layer $U_i$ can be written as:

$$In(U_i) = \text{Concat}(Out(U_{i-1}), \; Out(D_i)) \tag{10}$$

The noise prediction model takes three input: time step $t$, current latent-space representation $z_t$ and the condition embedding vector $c$. For time step $t$, we build a trainable embedding table to fetch time embedding and then add to $z_t$. The $t$-th row of time embedding table is initialized as:

$$\text{time emb} = \text{Concat}\left(\left\{\sin\left(t \cdot e^{-\frac{10i}{d/2-1}}\right)\right\}_{i=0}^{\frac{d}{2}-1}, \; \left\{\cos\left(t \cdot e^{-\frac{10i}{d/2-1}}\right)\right\}_{i=0}^{\frac{d}{2}-1}\right) \tag{11}$$

where $d$ is the dimension of embedding length, and is assigned to 64 in our model.

For condition embedding vector $c$, it is embraced in the cross attention block [Vaswani et al. 2017] in both sampling block and bottleneck block. Moreover, we deploy the self attention block [Vaswani et al. 2017] to consider the global details in latent vector, which promotes the consistency in QRS complex amplitude of generated ECG waveform.

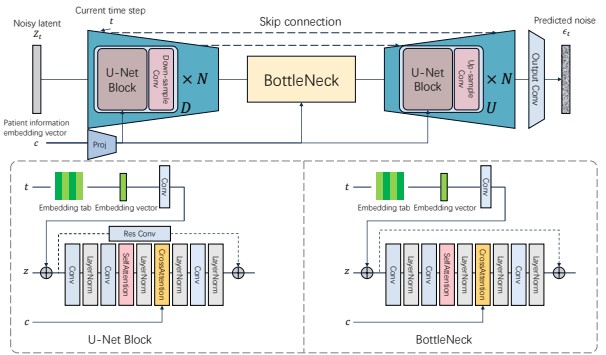

Figure 6: The detail architecture of noise predictor model in DiffuSETS.

## A.3 Processing of clinical text reports in DiffuSETS

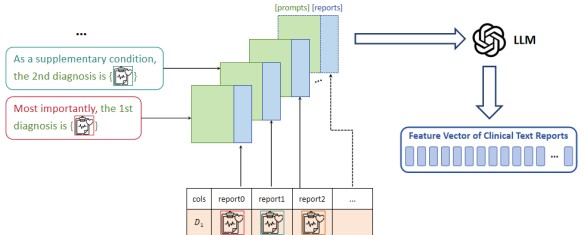

Figure 7: Processing of clinical text reports in DiffuSETS.

## A.4 Processing of patient-specific information in DiffuSETS

*A.4.1 Categorical Demographic Condition.* We categorize discrete, categorical conditions such as sex and race as categorical demographic conditions. This type of data can be processed to generate embedding vectors using a learnable classifier, or results can be directly obtained using a large language model. In our DiffuSETS method, we use the feature of sex to represent categorical demographic conditions in experiments. Since sex is binary data, it can be represented simply using 0 or 1.

*A.4.2 Numerical Demographic Condition.* We categorize continuous, numerical conditions such as age and weight as numerical demographic conditions. These types of data are stored in tables in numerical form. During the training and inference processes of the model, they can be directly utilized. In our DiffuSETS method, we use the feature of age to represent numerical demographic conditions in experiments. For this category of conditions, it is important to consider the data distribution and the removal of outliers.

*A.4.3 Other Health Condition.* Specifically, we categorize data related to patient health metrics such as heart rate and left ventricular ejection fraction (LVEF) as other health conditions. They can also affect the morphology of the ECG. Many of these types of data are recorded in dataset tables, and others require processing to be obtained. Notably, when an ECG is provided, these values can often be calculated. Therefore, in the task of generating ECGs, we can perform calculations on the generated ECGs to intuitively assess the generation effectiveness. In our DiffuSETS method, we use heart rate to represent other health conditions in experiments and have conducted

feature-level evaluation and analysis of this characteristic after generating the ECG, making full use of the data's intrinsic properties.

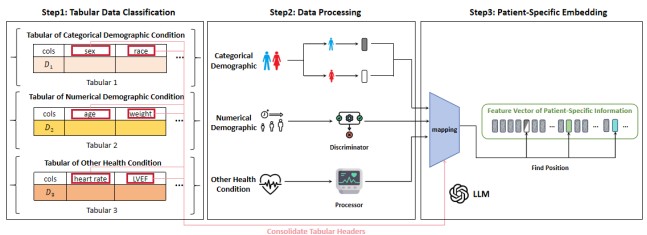

Figure 8: Processing of patient-specific information in DiffuSETS.

## B SUPPLEMENTAL EXPERIMENT

Besides the experiment we mentioned in Section 4.2.2 We also try to generate ECGs condition on different given heart rate feature while fixing other conditions. Specifically, we choose the clinical text report to the normal 'sinus rhythm', age to 50 and gender to female. Then we change the value of heart rate from 30 bpm to 150 bpm to test the mean and standard variance of generated ECG heart rate. The result are shown in Table 4

Also, we perform the diagnostic level test on PTB-XL dataset, and the result is shown in Table 5

Table 4: Feature level test result by DiffuSETS and ablation models.

| DiffuSETS Model* | Heart Rate Under Given Condition (Mean & Std) | | | | |
| | 30 bpm | 60 bpm | 90 bpm | 120 bpm | 150 bpm |
| --- | --- | --- | --- | --- | --- |
| DiffuSETS | 46.78 | 58.43 | 89.30 | 121.94 | 149.69 |
| | 9.23 | 5.13 | 0.58 | 3.74 | 8.54 |
| w/o PS-info† | - | - | - | - | - |
| | - | - | - | - | - |
| w/o VAE | 62.27 | 61.04 | 90.08 | 123.68 | 135.12 |
| | 11.39 | 6.30 | 2.95 | 4.97 | 14.37 |
| w/o PS-info&VAE† | - | - | - | - | - |
| | - | - | - | - | - |

\* Code names of DiffuSETS models are the same with those in Table 1
† No Patient-specific model cannot be specified with heart rate features.

Table 5: Diagnostic test result by DiffuSETS and ablation models on PTB-XL dataset.

| DiffuSETS Model* | CLIP Score | Clip Score Ground Truth | rCLIP Score | rFID Score | Precision | Recall | F1 Score |
| --- | --- | --- | --- | --- | --- | --- | --- |
| DiffuSETS | **0.27** | 0.60 | **0.45** | 2.15 | **0.88** | 0.50 | **0.64** |
| w/o PS-info | 0.23 | 0.60 | 0.39 | 2.95 | 0.61 | 0.32 | 0.42 |
| w/o VAE | 0.18 | 0.60 | 0.31 | 3.05 | 0.84 | 0.30 | 0.44 |
| w/o PS-info&VAE | 0.14 | 0.60 | 0.23 | 3.86 | 0.51 | **0.62** | 0.56 |

\* Code names of DiffuSETS models are the same with those in Table 1

## C EXPERT EVALUATION SAMPLE LIST

**Table 6: List of conditions for ECG generation in semantic alignment test of expert evaluation. Part I**

| Image No. | age | gender | heart rate | clinical text report |
|---|---|---|---|---|
| 0 | 55 | F | 59 | abnormal ecg. |
| 1 | 50 | M | 88 | atrial premature complexes. |
| 2 | 75 | M | 114 | incomplete rbbb. |
| 3 | 80 | M | 59 | anterolateral st-t changes may be due to hypertrophy and/or ischemia. |
| 4 | 58 | F | 98 | extensive st-t changes may be due to myocardial ischemia. |
| 5 | 76 | M | 53 | sinus bradycardia with 1st degree a-v block. |
| 6 | 55 | F | 78 | sinus rhythm. |
| 7 | 91 | M | 63 | sinus arrhythmia with borderline 1st degree a-v block. |
| 8 | 61 | M | 79 | repol abnrm suggests ischemia, lateral leads. |
| 9 | 83 | F | 107 | left axis deviation. |
| 10 | 91 | M | 71 | sinus rhythm with pvc(s) with pac(s). |
| 11 | 31 | M | 56 | inferior and anterior t wave changes are abnormal. |
| 12 | 83 | F | 61 | possible anterior infarct - age undetermined. |
| 13 | 74 | F | 48 | st junctional depression is nonspecific. |
| 14 | 22 | F | 91 | normal ecg. |
| 15 | 62 | M | 60 | low qrs voltages in precordial leads. |
| 16 | 75 | M | 52 | poor r wave progression - probable normal variant. |
| 17 | 58 | F | 66 | prolonged qt interval. |
| 18 | 73 | M | 56 | abnormal t, probable ischemia, lateral leads. |
| 19 | 89 | F | 52 | possible ectopic atrial bradycardia. |
| 20 | 57 | M | 71 | sinus rhythm with borderline 1st degree a-v block. |
| 21 | 83 | F | 72 | intraventricular conduction defect. |
| 22 | 85 | M | 79 | lateral st elevation - cannot rule out myocardial injury. |
| 23 | 47 | F | 96 | lateral st-t changes are nonspecific. |
| 24 | 38 | M | 73 | borderline ecg. |
| 25 | 55 | M | 135 | septal t wave changes are nonspecific. |
| 26 | 31 | M | 125 | short pr interval. |
| 27 | 58 | M | 70 | nonspecific t abnormalities, lateral leads. |
| 28 | 65 | F | 60 | left bundle branch block. |
| 29 | 70 | F | 59 | rightward axis. |
| 30 | 69 | M | 121 | atrial flutter. |
| 31 | 72 | M | 67 | leftward axis. |
| 32 | 85 | M | 74 | lateral t wave changes are probably due to ventricular hypertrophy. |
| 33 | 88 | F | 123 | probable accelerated junctional rhythm. |
| 34 | 68 | F | 41 | inferior/lateral st-t changes may be due to hypertrophy and/or ischemia. |
| 35 | 82 | F | 92 | possible atrial flutter. |
| 36 | 77 | F | 84 | qrs changes v3/v4 may be due to lvh but cannot rule out anterior infarct. |
| 37 | 72 | F | 76 | i.v. conduction defect. |
| 38 | 25 | F | 87 | lateral st changes are nonspecific. |
| 39 | 75 | F | 74 | lateral t wave changes are nonspecific. |
| 40 | 68 | F | 56 | probable old inferior infarct. |
| 41 | 55 | M | 87 | inferior infarct - age undetermined. |
| 42 | 55 | M | 73 | possible inferior infarct - age undetermined. |
| 43 | 89 | M | 87 | atrial fibrillation. |
| 44 | 71 | F | 81 | rsr'(v1) - probable normal variant. |
| 45 | 58 | M | 97 | - premature ventricular contractions. |
| 46 | 70 | F | 56 | normal ecg except for rate. |
| 47 | 53 | F | 106 | ventricular pacing. |
| 48 | 83 | F | 58 | irregular ectopic atrial bradycardia. |
| 49 | 78 | F | 69 | pacemaker rhythm - no further analysis. |

**Table 7: List of conditions for ECG generation in semantic alignment test of expert evaluation. Part II**

| Image No. | age | gender | heart rate | clinical text report |
|---|---|---|---|---|
| 50 | 74 | M | 79 | possible anteroseptal infarct - age undetermined. |
| 51 | 89 | M | 69 | right bundle branch block. |
| 52 | 21 | F | 63 | sinus rhythm with pac(s). |
| 53 | 67 | M | 69 | *** consider acute st elevation mi ***. |
| 54 | 20 | M | 98 | sinus tachycardia. |
| 55 | 91 | F | 69 | probable left atrial enlargement. |
| 56 | 82 | F | 103 | lateral st-t changes may be due to myocardial ischemia. |
| 57 | 74 | F | 105 | atrial fibrillation with rapid ventricular response. |
| 58 | 60 | M | 88 | inferior t wave changes are nonspecific. |
| 59 | 44 | F | 62 | inferior st-t changes are nonspecific. |
| 60 | 78 | M | 72 | abnormal r-wave progression, early transition. |
| 61 | 39 | F | 72 | st elev, probable normal early repol pattern. |
| 62 | 61 | F | 87 | — recording unsuitable for analysis - please repeat —. |
| 63 | 56 | F | 55 | atrial fibrillation with slow ventricular response. |
| 64 | 53 | F | 103 | possible biatrial enlargement. |
| 65 | 70 | F | 73 | inferior infarct, old. |
| 66 | 34 | m | 115 | anteroseptal t wave changes are nonspecific. |
| 67 | 73 | F | 70 | inferior/lateral st-t changes are probably due to ventricular hypertrophy. |
| 68 | 73 | F | 69 | possible left atrial abnormality. |
| 69 | 83 | M | 74 | iv conduction defect. |
| 70 | 87 | F | 70 | - premature ventricular contractions or aberrant ventricular conduction. |
| 71 | 50 | F | 85 | poor r wave progression v2-v4. |
| 72 | 59 | M | 62 | nonspecific t abnormalities, anterior leads. |
| 73 | 80 | M | 65 | demand pacing. |
| 74 | 70 | M | 97 | rbbb with left anterior fascicular block. |
| 75 | 58 | F | 82 | possible right atrial abnormality. |
| 76 | 61 | M | 44 | sinus bradycardia with sinus arrhythmia with 1st degree a-v block. |
| 77 | 82 | M | 57 | extensive infarct - age undetermined. |
| 78 | 82 | M | 65 | dual chamber pacemaker. |
| 79 | 60 | M | 64 | anteroseptal infarct - age undetermined. |
| 80 | 57 | F | 67 | anterior t wave changes are nonspecific. |
| 81 | 54 | F | 56 | sinus bradycardia with borderline 1st degree a-v block. |
| 82 | 78 | M | 56 | atrial fibrillation with slow ventricular response with paroxysmal idioventricular rhythm. |
| 83 | 62 | F | 79 | extensive st-t changes may be due to hypertrophy and/or ischemia. |
| 84 | 51 | M | 83 | septal and lateral st-t changes are nonspecific. |
| 85 | 91 | F | 100 | possible atrial flutter with rapid ventricular response. |
| 86 | 51 | F | 47 | - paroxysmal idioventricular rhythm or aberrant ventricular conduction. |
| 87 | 63 | M | 80 | rbbb and lpfb. |
| 88 | 66 | F | 83 | t wave changes in lateral leads. |
| 89 | 74 | F | 132 | these minor changes are of equivocal significance only. |
| 90 | 75 | F | 52 | inferior/lateral t changes are probably due to ventricular hypertrophy. |
| 91 | 55 | M | 105 | — suspect arm lead reversal - only avf, v1-v6 analyzed —. |
| 92 | 84 | F | 100 | lvh with secondary repolarization abnormality. |
| 93 | 64 | F | 86 | nonspecific repol abnormality, diffuse leads. |
| 94 | 18 | M | 84 | low voltage, precordial leads. |
| 95 | 80 | M | 88 | nonspecific t abnrm, anterolateral leads. |
| 96 | 70 | F | 51 | extensive st-t changes are nonspecific. |
| 97 | 65 | F | 47 | sinus bradycardia. |
| 98 | 82 | F | 69 | a-v sequential pacemaker. |
| 99 | 62 | F | 103 | low qrs voltages in limb leads. |

# D    MORE EXAMPLES OF GENERATED ECG

Here we list more generated ECG data. These results show that our model can generate ECGs properly reflecting input clinical text report no matter from general view or from single beat view.

## D.1    Mostly normal ECG. The diagnosis text of the following ECG will be reflected at the overall level.

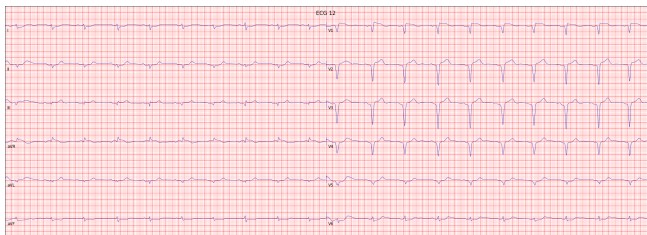

Figure 9: Age: 55; Heart Rate: 59; Text: abnormal ecg.

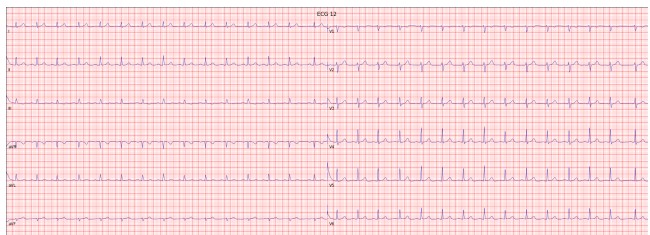

Figure 10: Age: 22; Heart Rate: 91; Text: normal ecg.

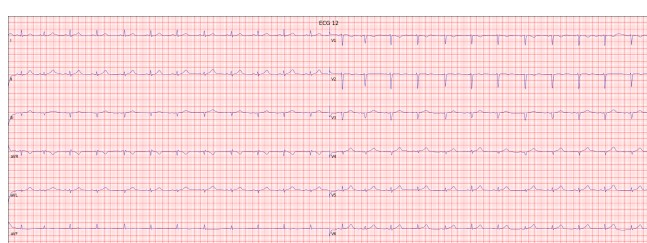

Figure 11: Age: 38; Heart Rate: 73; Text: borderline ecg.

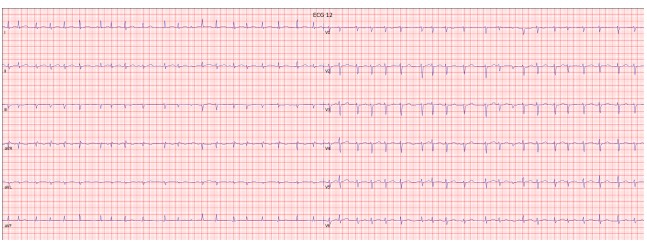

Figure 12: Age: 74; Heart Rate: 132; Text: these minor changes are of equivocal significance only.

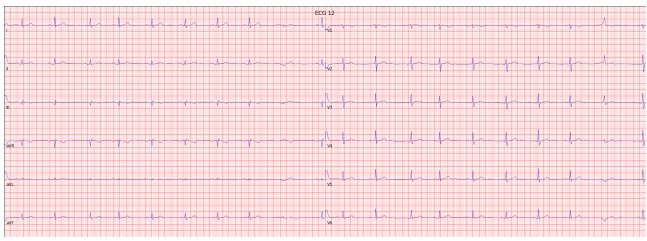

Figure 13: Age: 70; Heart Rate: 56; Text: normal ecg except for rate.

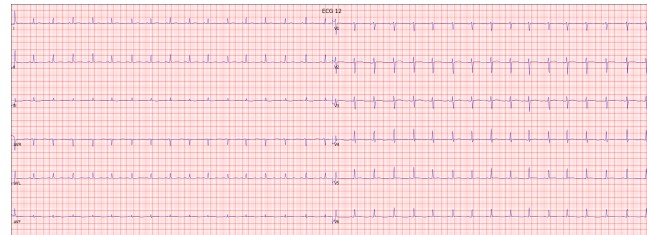

Figure 14: Age: 20; Heart Rate: 98; Text: sinus tachycardia.

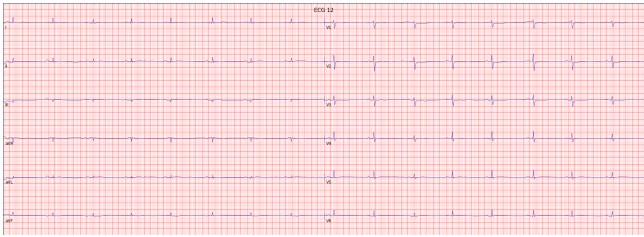

Figure 15: Age: 65; Heart Rate: 47; Text: sinus bradycardia.

## D.2    Pacemaker ECG. The diagnosis text of the following ECG will be reflected at the overall level.

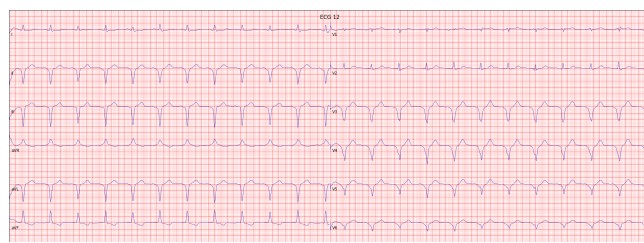

Figure 16: Age: 78; Heart Rate: 69; Text: pacemaker rhythm - no further analysis.

## D.3    Axis abnormal ECG. The diagnosis text of the following ECG can be reflected from some leads together.

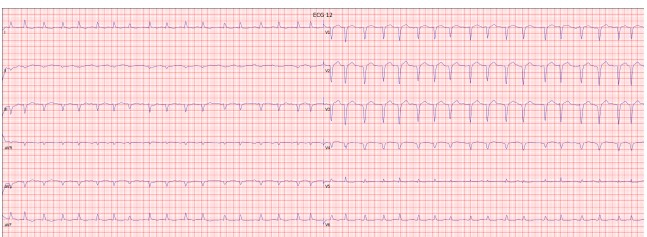

Figure 17: Age: 83; Heart Rate: 107; Text: left axis deviation.

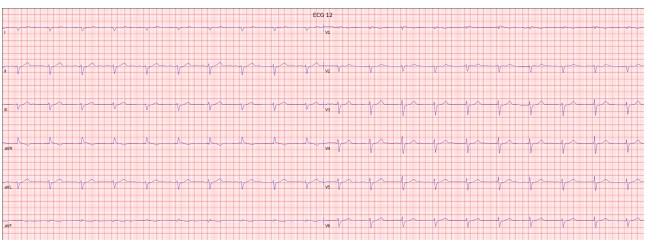

Figure 18: Age: 70; Heart Rate: 59; Text: rightward axis.

## D.4 Conduction abnormal ECG. The diagnosis text of the following ECG can be reflected from every beat level.

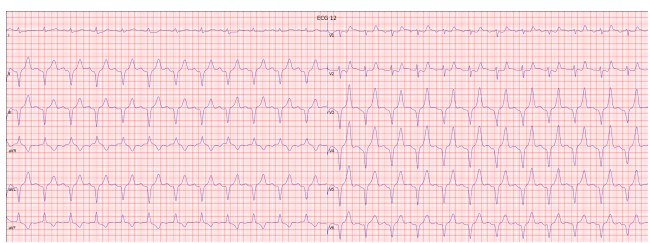

Figure 19: Age: 83; Heart Rate: 74; Text: iv conduction defect.

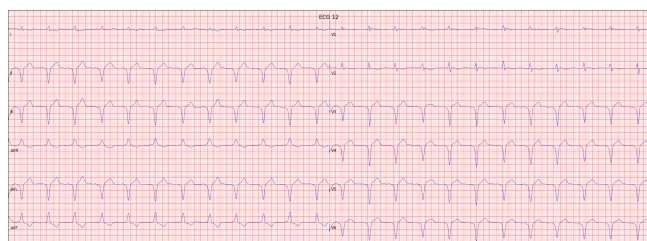

Figure 20: Age: 83; Heart Rate: 72; Text: intraventricular conduction defect.

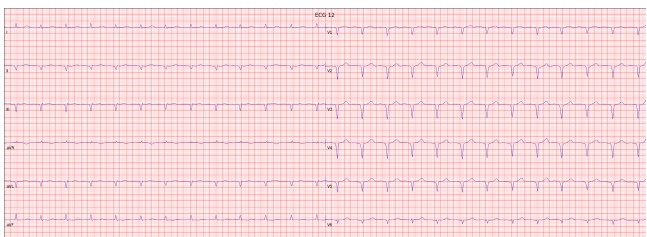

Figure 21: Age: 72; Heart Rate: 76; Text: i.v. conduction defect.

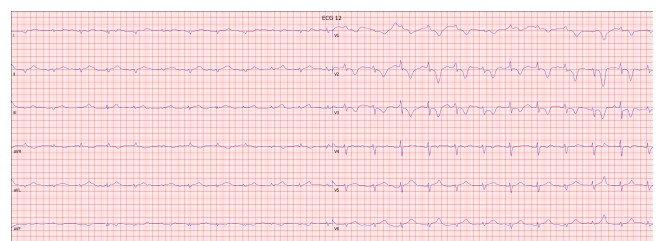

Figure 22: Age: 89; Heart Rate: 69; Text: right bundle branch block.

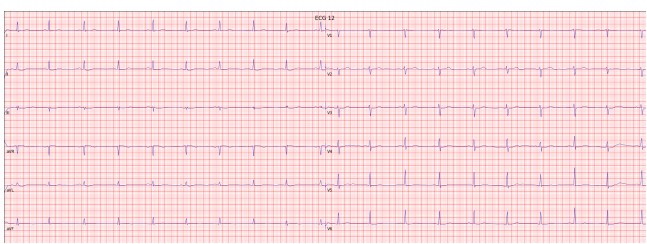

Figure 23: Age: 54; Heart Rate: 56; Text: sinus bradycardia with borderline 1st degree a-v block.

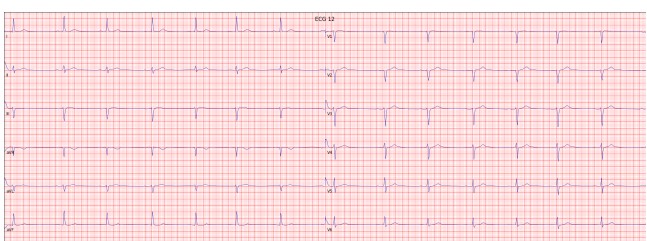

Figure 24: Age: 61; Heart Rate: 44; Text: sinus bradycardia with sinus arrhythmia with 1st degree a-v block.

## D.5 Cardiac structure abnormal ECG. The diagnosis text of the following ECG can be reflected from every P wave.

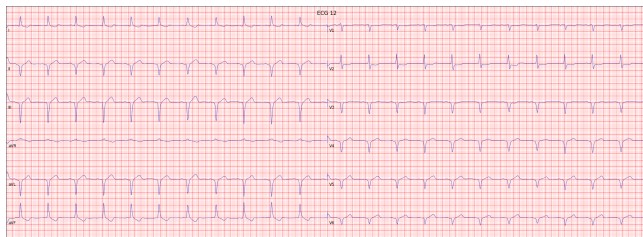

Figure 25: Age: 73; Heart Rate: 69; Text: possible left atrial abnormality.

### D.6 Premature contractions ECG. The diagnosis text of the following ECG can be reflected from a single occasional beat. Multiple abnormalities are also present.

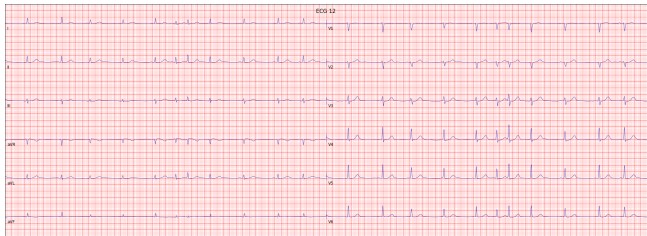

Figure 26: Age: 21; Heart Rate: 63; Text: sinus rhythm with pac(s).

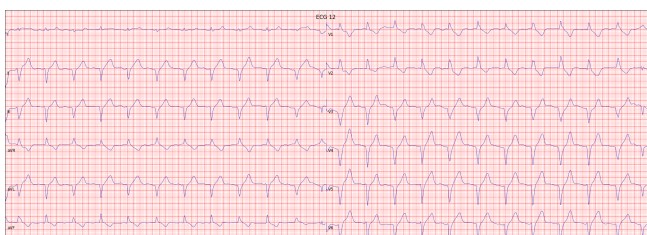

Figure 27: Age: 87; Heart Rate: 70; Text: - premature ventricular contractions or aberrant ventricular conduction.

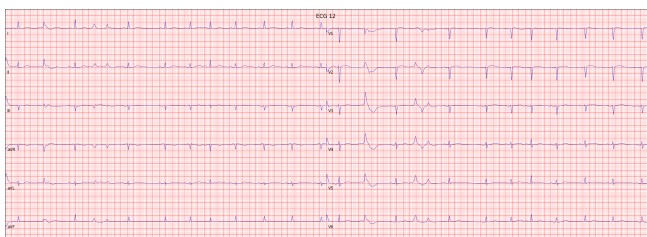

Figure 28: Age: 91; Heart Rate: 71; Text: sinus rhythm with pvc(s) with pac(s).

### D.7 Atrial flutter and atrial fibrillation ECG. The diagnosis text of the following ECG can be reflected in rhythm level. Every beats are slightly abnormal. Multiple abnormalities are also present.

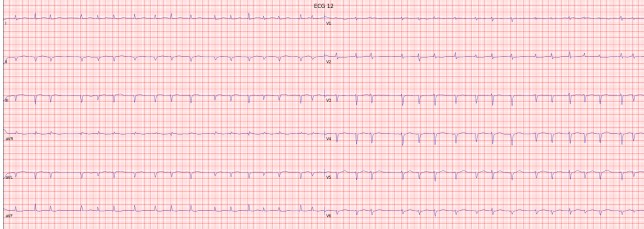

Figure 29: Age: 91; Heart Rate: 100; Text: possible atrial flutter with rapid ventricular response.

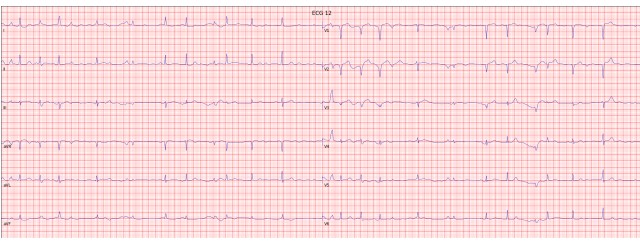

Figure 30: Age: 74; Heart Rate: 105; Text: atrial fibrillation with rapid ventricular response.

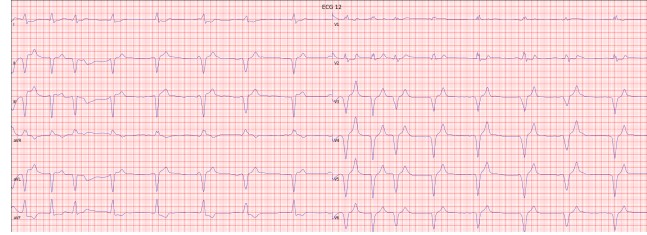

Figure 31: Age: 78; Heart Rate: 56; Text: atrial fibrillation with slow ventricular response with paroxysmal idioventricular rhythm.

### D.8 Possible myocardial ischemia ECG. Very dangerous. The diagnosis text of the following ECG can be reflected in every beat, but only specific lead(s) might present this change.

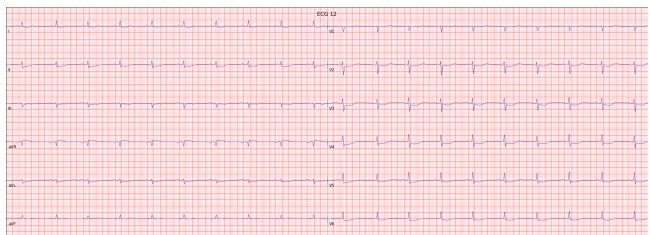

**Figure 32: Age: 80; Heart Rate: 59; Text: anterolateral st-t changes may be due to hypertrophy and/or ischemia.**

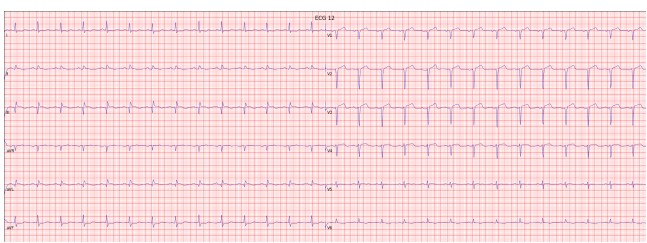

**Figure 36: Age: 77; Heart Rate: 84; Text: qrs changes v3/v4 may be due to lvh but cannot rule out anterior infarct.**

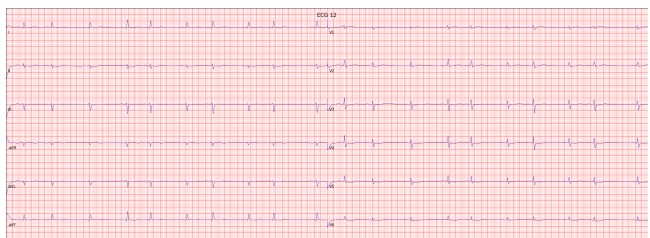

**Figure 33: Age: 73; Heart Rate: 56; Text: abnormal t, probable ischemia, lateral leads.**

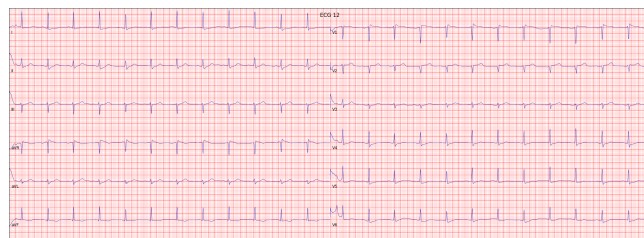

**Figure 37: Age: 75; Heart Rate: 74; Text: lateral t wave changes are nonspecific.**

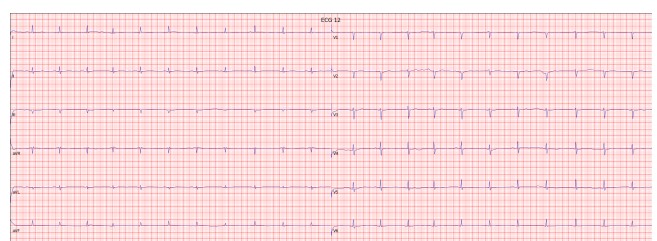

**Figure 34: Age: 58; Heart Rate: 70; Text: nonspecific t abnormalities, lateral leads.**

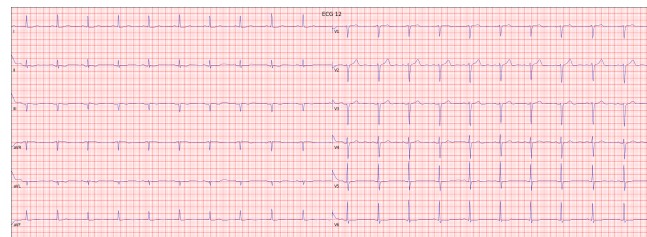

**Figure 38: Age: 44; Heart Rate: 62; Text: inferior st-t changes are nonspecific.**

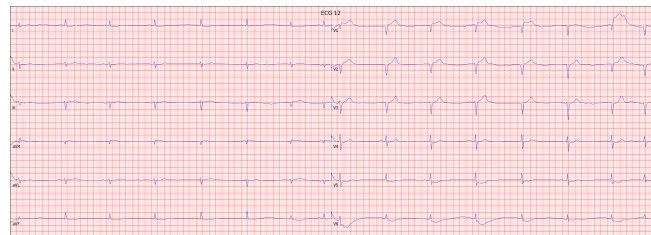

**Figure 35: Age: 68; Heart Rate: 41; Text: inferior/lateral st-t changes may be due to hypertrophy and/or ischemia.**

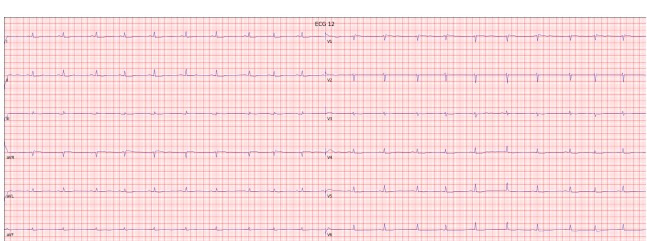

**Figure 39: Age: 59; Heart Rate: 62; Text: nonspecific t abnormalities, anterior leads.**

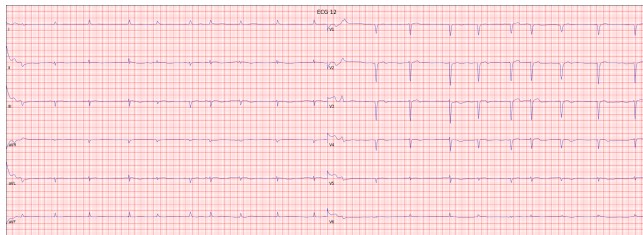

**Figure 40: Age: 82; Heart Rate: 57; Text: extensive infarct - age undetermined.**

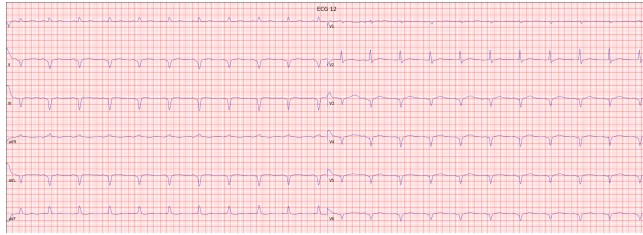

**Figure 41: Age: 60; Heart Rate: 64; Text: anteroseptal infarct - age undetermined.**

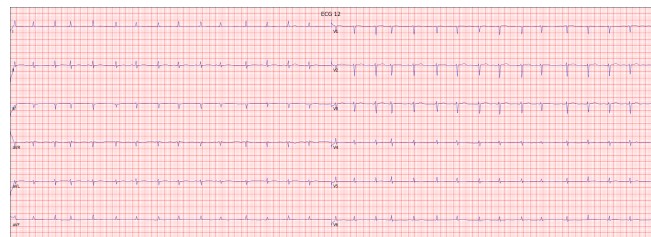

**Figure 42: Age: 80; Heart Rate: 88; Text: nonspecific t abnrm, anterolateral leads.**

### D.9 Low voltages ECG. The diagnosis text of the following ECG can be reflected at the overall level, but only specific leads.

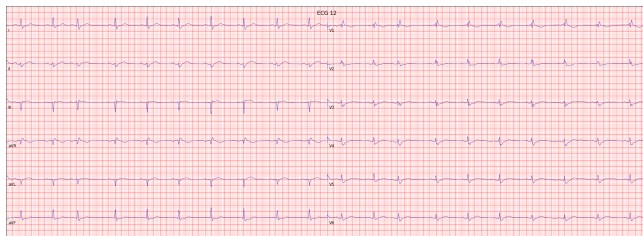

Figure 43: Age: 62; Heart Rate: 60; Text: low qrs voltages in precordial leads.

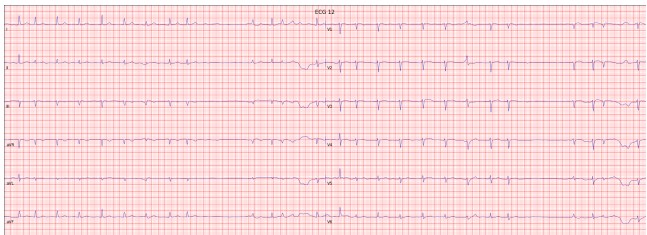

Figure 44: Age: 18; Heart Rate: 84; Text: low voltage, precordial leads.

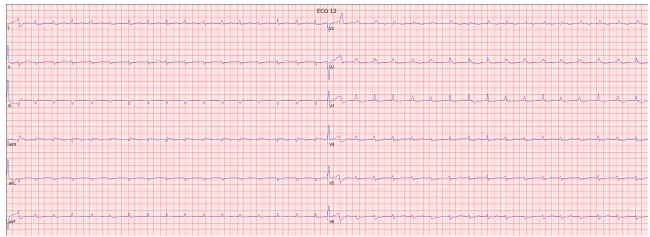

Figure 45: Age: 62; Heart Rate: 103; Text: low qrs voltages in limb leads.