# OpenReview forum: "DiffuSETS: 12-lead ECG Generation Conditioned on Clinical Text Reports and Patient-Specific Information"
_KDD.org/2024/Workshop/AIDSH — KDD-AIDSH 2024 Poster_

### Official Review · Reviewer_qof2 · 2024-06-12

**Rating:** 8
**Confidence:** 5

**Review:**

This paper is intriguing and the method is reasonable. The approach of using the LLM embedding layer to process patient personalized information is interesting and makes sense. The validation is comprehensive and convincing.

There are only a few minor issues:

Figure 4 should be a figure and a table, not just a figure caption.

Tables 1, 2, and 4 in Section 4.5 should be correctly referenced as Tables 1, 2, and 4.

I vote for a clear accept for this paper.

---

### Official Review · Reviewer_5Ss9 · 2024-06-15
**DiffuSETS: 12-lead ECG Generation Conditioned on Clinical Text Reports and Patient-Specific Information**

**Rating:** 5
**Confidence:** 3

**Review:**

Summary：

In this paper, the authors propose a novel method for generating ECG waveforms using a diffusion model, which specifies the desired features of ECG waveform generation through natural language input. This method, called DiffuSETS, can generate corresponding ECG waveforms by inputting a patient's age, gender, heart rate, and clinical text. Specifically, DiffuSETS controls the generation outcome by incorporating text embeddings produced by an LLM and age and gender information processed by an LLM as conditions during the Denoising process. The results provided by the authors look very promising and can be described as a very direct and effective method. However, I believe using natural language as input to generate an ECG model has limited clinical significance; clinicians might prefer to generate diagnostic language directly through ECG. Moreover, the potential for retraining with the generated data is not yet clear, and this paper's contribution to the community may be very limited. If the authors could continue to discuss the application scenarios of this paper, it would increase the persuasiveness of the article.

Strengths:

- The article is well-written.
- Using experts to evaluate the quality of the generated samples is convincing.
- The article has a large number of generated results, which are sufficient to demonstrate the effectiveness of the proposed method.

Weaknesses:

- The motivation for using natural language to generate ECG waveforms is weak.
- From the results given in the appendix, it can be seen that the model performs poorly in generating heart rates.
- MAE might not be a good evaluation metric for generation tasks.

Questions:

- Would it be more efficient to use an LLM to directly extract important information from diagnostic texts as tags to input into the model, compared to directly computing text embeddings? For example, if a patient's clinical text is 'atrial fibrillation with slow ventricular response,' then the patient's characteristics would only include atrial fibrillation and slow ventricular response, and these could be encoded using a one-hot method. Intuitively, before better performing LLMs are available, explicitly encoding patient features could provide stronger conditions for the model.
- Using MAE to assess the generative performance of the model might not be a good choice, especially in cases where there is a phase difference between the generated results and the original waveform. Even if the model can accurately reproduce the patient's features, MAE could be misleading due to differences in the timing of the disease's appearance or initial phase differences.
- For expert testing, I believe a more effective testing method would be to have experienced experts diagnose all samples that include some generated samples and calculate the alignment of all diagnostic results, rather than directly asking experts to try to align the semantic information of the generated samples. This is because experts may not guarantee complete accuracy, and it is important not to let the experts have prior knowledge of the samples, which could lead to biased perspectives.

---

### Decision · Program_Chairs · 2024-06-28

Accept (Poster)